# Design of Variable Stiffness Trajectories with Cubic Ferguson Curve

**DOI:** 10.3390/ma16216866

**Published:** 2023-10-26

**Authors:** Deli Zhang, Kai Wang, Xiaoping Wang

**Affiliations:** College of Mechanical and Electrical Engineering, Nanjing University of Aeronautics and Astronautics, Nanjing 210016, China; nuaazdl@126.com (D.Z.); wang_kai@nuaa.edu.cn (K.W.)

**Keywords:** variable stiffness, full cover, automatic fiber placement, buckling load

## Abstract

To design a class of full cover trajectories that satisfy curvature limitations and enhance the buckling load of constructed laminates, a variable stiffness laminate is proposed by applying the cubic Ferguson curve. First, the traditional explicit form of the cubic Ferguson curve is redefined as polar coordinates, two connected Ferguson curve segments with three extra parameters are applied to describe full cover trajectories, and the effects on trajectories introduced by these modifications are discussed. Then, the finite element method is used to introduce parameters for analyzing the buckling load of the designed variable stiffness laminates. Numerical experiments show that automatic fiber placement (AFP) trajectories described by the cubic Ferguson curve can automatically reach $C^{1}$ continuity and can be locally modified by adjusting the introduced parameters. Compared with traditional constant stiffness laminates, the variable stiffness laminates designed using the proposed method exhibit a higher buckling load and better stability.

## 1. Introduction

Compared with traditional materials, fiber-reinforced composites have many advantages, including higher strength-to-weight and stiffness-to-weight ratios, resistance to corrosion, and ease of shaping and tailoring their structural configurations. Therefore, they are widely applied in advanced high-technology fields, such as aeronautics, transportation, and electrical power, in areas including aircraft structural components, artificial satellites, and high-speed rail cars. At present, automatic placement technology has become the main processing method for high-performance composite materials and is widely applied in modern aircraft manufacturing enterprises. The technology comprises filament winding, tape placement, and automatic fiber placement (AFP). The AFP can independently operate each prepreg tow, and the width of the prepreg assembled can be adjusted based on the surface features and manufacturing requirements. This approach has a broader applicability than filament winding, and tape placement can be placed in a geodesic manner. Therefore, this technology is particularly suitable for the automatic formation of large curvatures and complex composite components, such as S-shaped inlets and load bearings.

As an automatic as well as the most-advanced manufacturing technique for producing fiber-reinforced composite components, efficient and reasonable trajectory generation tactics are key factors for determining the quality and efficiency of fiber placement. The existing approaches to trajectory generation for AFP can be classified into constant stiffness and variable stiffness methods. If the ply structures of these components are the same in all positions, then the stiffness is the same, and they are named constant stiffness components. Several studies have been conducted on this topic: [1,2,3,4,5,6,7]. However, in practice, these components have varying stiffness, which improves the flexibility of AFP trajectory generation and avoids stress concentration problems. A range of experiments and studies have illustrated that curved fibers can further enhance the structure of components in terms of strength, buckling capacity [8,9,10], post-buckling capacity [11,12,13,14,15], and dynamic features [16]. Based on these, various studies have analyzed and modeled variable stiffness methods.

Wang defined three manufacturing constraints and optimized the fiber directions [2]. Based on the finite element method, Cho obtained the optimal fiber direction over elements [17]. Bruyneel proposed a gradient-based method of moving asymptotes and a method of moving asymptotes that addresses the structural weight, stiffness, and strength, and optimizes the fiber direction [18]. Setoodth replaced the geometry factor with a fiber angle to define the variable stiffness laminates [19]. Aiming at conical shells, Blom defined the geodesic path, constant angle path, paths with linearly varying angles, and constant curvature paths [20]. Furthermore, the author applied the maximum fundamental eigenfrequency to design variable stiffness paths [21]. Based on the studies conducted by Blom, Fayazbakhsh defined a new fiber direction using the sine function and showed that this method can capture the effects of gaps and overlaps in variable stiffness laminates [22]. Similarly, Nik used the cos function to define the fiber direction, thereby enabling the tracing of a family of variable stiffness paths [23]. Using a linear system, Riuhi illustrated the design, manufacturing, and testing procedures for a variable stiffness composite cylinder [24]. In addition to the aforementioned linearly variable cases, various nonlinear models have also been developed. Alhajahmad applied Lobatto–Legendre functions to describe the initial path and compared the bulking analysis with Ritz’s method [25]. Wu presented a method for defining variable stiffness paths using Lagrange polynomials and optimized the laminates using the Rayleigh–Ritz method [26]. Kim utilized the B*é*zier curve to define the AFP trajectories and showed that this method can avoid the problem of caps and overlaps [27]. For more examples, see [28,29,30,31,32,33].

In this study, our main contribution is that we provide a novel scheme using the cubic Ferguson curve to design variable stiffness laminates. The proposed method adjusts the shape of the generated trajectories, reduces their curvature, and enhances the buckling load by locally modifying the introduced parameters. Compared with traditional constant stiffness laminates, the proposed variable stiffness laminate performs better in terms of the buckling load, and the proposed method has several advantages, such as simple expression, $C^{1}$ continuity and aligning with tangents of the trajectories, thus, providing a new pipeline for variable laminates. The rest of this paper is organized as follows. The definition of variable stiffness laminates is presented in Section 2. The buckling loads of the constant and variable laminates are discussed in Section 3. Conclusions are presented in Section 4.

## 2. Definition of Variable Stiffness Laminates

### 2.1. Cubic Ferguson Curve

For AFP technology, the basis of assumption relative to full cover trajectories is that given an initial trajectory, the next must offset along a special direction up to a given width (i.e., fiber width *d*) over the mold surface. Therefore, there must be an initial trajectory and controlling its angle allows a class of variable stiffness trajectories to be obtained that can fully cover the mold surface.

The principle of defining AFP trajectories using cubic Ferguson curves with two parameters $\alpha_{1}$ and $\alpha_{2}$ is illustrated in Figure 1. The cubic Ferguson curve usually has a simple expression that includes only two endpoints $P_{1}$ and $P_{2}$ and two related direction vectors $T_{1}$ and $T_{2}$, and these trajectories can automatically achieve $C^{1}$ continuity at their endpoints [34]. Most importantly, the inherent vectors $T_{1}$ and $T_{2}$ align with the tangents of the trajectories. This practical property can be utilized to produce either constant or variable stiffness trajectories. To better illustrate this, we first redefined the Ferguson curve using polar coordinates for parameter *t*:(1)$$\begin{array}{c} Q\left(t\right)=F_{0}\cdot T_{0}+G_{0}T_{1}+G_{1}T_{2}+F_{1}T_{3},\\ T_{0}=\left[0,0\right],\\ T_{1}=\alpha_{1}\lambda \left[1,\tan\theta_{1}\right],\\ T_{2}=\alpha_{2}\left(1-\lambda \right)\left[1,\tan\theta_{2}\right],\\ T_{3}=h\left[\left(1-\lambda \right)/\tan\theta_{2}+\lambda /\tan\theta_{1},1\right], \end{array}$$
where $\theta_{1}$ and $\theta_{2}$ are the included angles between the vectors $T_{1}$ and $T_{2}$ and the *x*-axis, respectively. The extra introduced vectors $T_{0}$ and $T_{3}$ are determined by the vectors $\vec{P_{0}P_{0}}$ and $\vec{P_{0}P_{1}}$. It is noteworthy that the two vectors must not be collinear, and they must have an interpolating point $P^{*}$ in the first quadrant. In this case, the angles $\theta_{1}$ and $\theta_{2}$ must simultaneously satisfy $45^{\circ }<\theta_{1},\theta_{2}<90^{\circ }$ and $45^{\circ }<\theta_{2},\theta_{1}<90^{\circ }$. If the included angles are close to $\pm 90^{\circ }$, then one can take $T_{1}=\pm \lambda \alpha_{1}\left[0,1\right]$ and $T_{2}=\pm \left(1-\lambda \right)\alpha_{2}\left[0,1\right]$. Moreover, $\alpha_{1}$ and $\alpha_{2}$ are two inherent parameters that are usually assigned to vectors $T_{0}$ and $T_{1}$, respectively. Both parameters can be applied to modify the cubic Ferguson curve locally. The related cubic basis functions of the Ferguson curves are
(2)$$\left\{\begin{array}{c} F_{0}=2t^{3}-3t^{2}+1,\\ G_{0}=t{\left(t-1\right)}^{2},\\ G_{1}=t^{2}\left(t-1\right),\\ F_{1}=-2t^{3}+3t^{2}. \end{array}\right.$$
In this case, one can directly obtain the coordinates of the generated Ferguson curve segment Q(t) on the xoy-plane; it follows that
(3)$$x\left(t\right)=\alpha_{1}\cdot G_{0}+\alpha_{2}\cdot G_{1}+h\left(\frac{1-\lambda }{\tan(\theta_{2})}+\frac{\lambda }{\tan(\theta_{1})}\right)\cdot F_{1}$$
and
(4)$$y\left(t\right)=\alpha_{1}\cdot \tan\left(\theta_{1}\right)\cdot G_{0}+\alpha_{2}\cdot \tan\left(\theta_{2}\right)\cdot G_{1}+h\cdot F_{1}.$$
Furthermore, the included angle θ between the Ferguson curve and the x-axis is directly computed using the following rule:(5)$$\theta =\tan^{-1}\left(\frac{dy}{dt}\cdot \frac{dt}{dx}\right)=\frac{\alpha_{1}\cdot \tan\left(\theta_{1}\right)\cdot G_{0}^{\prime }+\alpha_{2}\cdot \tan\left(\theta_{2}\right)\cdot G_{1}^{\prime }+h\cdot F_{1}^{\prime }}{\alpha_{1}\cdot G_{0}^{\prime }+\alpha_{2}\cdot G_{1}^{\prime }+h\left(\frac{1-\lambda }{\tan(\theta_{2})}+\frac{\lambda }{\tan(\theta_{1})}\right)\cdot F_{1}^{\prime }}.$$
It is not difficult to see that the included angles between the AFP trajectories and *x*-axis are variables with parameter *t*.

### 2.2. Local Adjustable Property

The shape of a cubic Ferguson curve segment can be locally adjusted using the inherent parameters $\alpha_{1}$ and $\alpha_{2}$ and the newly introduced parameter λ. In general, the coordinates of interpolating point $P^{*}$ and end endpoint $P_{2}$ are affected by the parameter λ∈[0,1]. If λ is close to 1, the coordinate of interpolating point $P^{*}$ on *y*-axis is close to *h*. Otherwise, it is close to 0. Moreover, if λ is close to 1, the coordinate of interpolating point $P^{*}$ is close to the *x*-axis, as depicted in Figure 2. To avoid singularity, λ is always taken as a fixed value, i.e., λ=0.5, in the following discussion. However, in the case of $\alpha_{1}$ and $\alpha_{2}$, they can be observed as locally adjustable parameters, as depicted in Figure 3. Both parameters can be applied to modify the local shape of the generated trajectories and to smooth large curvatures. To obtain an ideal trajectory without gaps, overlaps, or wrinkles, the magnitudes of $\alpha_{1}$ and $\alpha_{2}$ cannot take a large value to avoid some singularities. The influence of parameters $\alpha_{1}$ and $\alpha_{2}$ on cubic Ferguson curves and the determination of the interval magnitude is not the main work of this paper and the related work can be found in [34,35]. Parameters $\alpha_{1}$ and $\alpha_{2}$ should not be greater than three times the distance between two endpoints $P_{1}$ and $P_{2}$ empirically, i.e., $\alpha_{1},\alpha_{2}<3||P_{2}-P_{1}\left|\right|$. Generally, the curvature formula for a given trajectory can be computed as follows:(6)$$\kappa =\frac{\left|x^{\prime }\left(t\right)y^{\prime \prime }\left(t\right)-x^{\prime \prime }\left(t\right)y^{\prime }\left(t\right)\right|}{{\left[x^{\prime }{\left(t\right)}^{2}+y^{\prime }{\left(t\right)}^{2}\right]}^{3/2}}.$$

### 2.3. Connection of AFP Trajectories

A single complete AFP trajectory always comprises multiple Ferguson curve segments. When the first segment is given, the start endpoint of the second segment coincides with the end endpoint of the first segment (i.e., $C^{0}$ continuity). To meet the AFP technology requirements, these trajectories must satisfy high-order continuity that can eliminate unnecessary cusps, loops, and inflection points. Fortunately, cubic Ferguson curves can automatically achieve $C^{1}$ continuity (i.e., $Q_{i}\left(1\right)=Q_{i+1}\left(0\right)$) when a series of fixed tangent vectors is provided at the given knots, as shown in Figure 4.

If a complete AFP trajectory comprises *N* Ferguson curve segments, it contains N+1 knots $P_{i}$ and the related tangent vector $T_{i}$(i=0,1,2,3,⋯). Moreover, it introduces 2N-adjusted parameters $\alpha_{1i}$ and $\alpha_{2i}$, which can be used to locally modify the defined trajectories. Then, a composite trajectory that can automatically achieve $C^{1}$ continuity is defined as follows:(7)$$Q\left(t\right)=\overset{N}{\underset{k=1}{⋃}}[Q_{i-1}\left(1\right)+Q_{i}\left(t\right)],t\in \left[0,1\right],$$
where $Q_{0}\left(1\right)=\left[0,0\right]$ and $Q_{k}\left(t\right)$ is the *k*-th curve segment.

Referring to the AFP trajectories defined by the B*é*zier curves, we use the following expression to describe the variable stiffness ply [27]:(8)$$\begin{array}{c} \Theta <\theta_{1}\left(\alpha_{1},\alpha_{2}\right)\theta_{2}\left(\alpha_{2},\alpha_{3}\right)\theta_{3},\cdots ,\\ \theta_{k}\left(\alpha_{k},\alpha_{k+1}\right)\theta_{k+1},\cdots ,\theta_{n-1}\left(\alpha_{n-1},\alpha_{n}\right)\theta_{n}>, \end{array}$$
where θ denotes the angle between the fiber and *x*-axis; $\theta_{k}$ and $\theta_{k+1}$ denote the start and end angles of the *k*-th trajectory segments, respectively; and $\alpha_{k}$ and $\alpha_{k+1}$ denote the local parameters of this segment, respectively. In this study, we further simplified the above definition and used two curve segments to describe the AFP trajectory with three local parameters $\alpha_{1},\alpha_{2},\alpha_{3}$ and three knots $P_{1},P_{2},P_{3}$. Then, the variable stiffness ply can be expressed as follows:(9)$$\Theta <\theta_{1}\left(\alpha_{1},\alpha_{2}\right)\theta_{2}\left(\alpha_{2},\alpha_{3}\right)\theta_{3}>.$$
Moreover, it is noteworthy that their related adjacent ply can be expressed as follows:(10)$$90^{\circ }-\Theta <90^{\circ }-\theta_{1}\left(\alpha_{1},\alpha_{2}\right)90^{\circ }-\theta_{2}\left(\alpha_{2},\alpha_{3}\right)90^{\circ }-\theta_{3}>.$$
The full cover trajectories of variable stiffness laminates can be traced by offsetting the initial trajectory expressed by the cubic Ferguson curves along the *x*-axis for a fixed fiber width *d*. In our cases, the initial trajectory is designed using two piecewise segments, and it is affected by three local parameters $\alpha_{1},\alpha_{2}$, and $\alpha_{3}$ and three direction angles $\theta_{1},\theta_{2}$, and $\theta_{3}$. Some adjustable cases of full cover trajectories based on different setting parameters are depicted in Figure 5. Some fiber directions on special elements over a variable composite laminate are shown in Figure 6. In particular, if one takes the triple direction angle as $\theta_{i}=\pm 45^{\circ },i=1,2,3$, then the generated full cover trajectories are equal to the straight fiber trajectories designed in the direction $\pm 45^{\circ }$. Generally, trajectories with a small magnitude of the local parameters $\alpha_{i}\left(i=1,2,3\right)$ are dense in the middle of the plate, as demonstrated in Figure 5a–f.

## 3. Analyses and Discussions

To demonstrate the efficiency of the proposed scheme, we numerically simulated 8-layer square laminates with a width of 150mm, applied a uniform force to the laminate’s right boundary, and performed compression analysis on the laminate. For this, we used Abaqus software (https://www.3ds.com/products-services/simulia/products/abaqus/). The essential boundary and Neiman boundary conditions are illustrated in Figure 7. Load *q* is 1. Moreover, the thickness and width of the applied materials are 0.125mm and 5mm, respectively. The other parameters are listed in Table 1.

First, we performed buckle analyses of the constant and variable stiffness. In this case, some optional settings of the fiber directions of the traditional constant stiffness are as follows: <$0^{\circ },90^{\circ },45^{\circ },-45^{\circ }$>, <$90^{\circ },0^{\circ },45^{\circ },-45^{\circ }$>, <$45^{\circ },-45^{\circ },0^{\circ },90^{\circ }$>, <$45^{\circ },-45^{\circ },90^{\circ },0^{\circ }$>, <$0^{\circ },45^{\circ },90^{\circ },-45^{\circ }$>, <$0^{\circ },-45^{\circ },90^{\circ },45^{\circ }$>, <$0^{\circ },45^{\circ },-45^{\circ },90^{\circ }$>, and <$0^{\circ },-45^{\circ },45^{\circ },90^{\circ }$>. However, for the variable laminates, the two adjacent variable stiffness plies are $\theta <\theta_{1}\left(\alpha_{1},\alpha_{2}\right)\theta_{2}\left(\alpha_{2},\alpha_{3}\right)\theta_{3}$> and π/2−θ. Detailed buckling analyses of constant and variable stiffness laminates are shown in Table 2.

It is not difficult to observe that the optimal fiber direction sequences on constant stiffness laminates are <$0^{\circ },90^{\circ },45^{\circ },-45^{\circ }$> and <$45^{\circ },-45^{\circ },0^{\circ },90^{\circ }$>. This is because their buckling loads can be up to 1215.8N. However, the variable stiffness laminates are not strictly better or worse than constant stiffness laminates. In fact, their buckling loads depend on the parameter settings (α and θ). Therefore, one can optimize the buckling load of variable stiffness laminates by modifying the parameters of the Ferguson curves. The effects of the parameter settings on variable stiffness laminates are presented in Table 2 and Figure 5.

To simplify the discussion, a symmetric initial trajectory is used in this study, that is, two kinds of parameters α and θ are taken as $\alpha_{1}=\alpha_{3}$ and $\theta_{1}=\theta_{3}$. If the other parameters are the same, a small α has an optimal buckling load, and one can observe that the optimal parameter sequence is $\alpha_{1}=\alpha_{3}=50$ and $\alpha_{2}=275$. However, for θ, $\theta_{2}$ close to $90^{\circ }$ and $\theta_{1}$ and $\theta_{3}$ close to 0 have a better buckling load, as shown in Figure 8. Reasonable and ideal parameter sequences according to the buckling analysis are <$30^{\circ }\left(50,275\right)90^{\circ }\left(275,50\right)30^{\circ }$>. In practice, another influencing factor affecting variable laminates is the curvature of the cubic Ferguson curve, and experiments were conducted to determine the curvature limitation that prevents placement fibers from becoming wrinkled, as shown in Table 2 and Figure 5. The curvatures of the cubic Ferguson curve with different parameter settings are shown in Figure 9. In this case, the optional optimal parameter sequences according to the trajectory curvature are <$26^{\circ }\left(60,275\right)90^{\circ }\left(275,150\right)30^{\circ }$>. It is not difficult to see that trajectories with a better buckling load may not necessarily have a small curvature, and vice versa. Therefore, a practical optimal parameter sequence is needed to consider both of curvature limitation and buckling load. Summarily, a class of optimal parameter sequences <$30^{\circ }\left(60,275\right)90^{\circ }\left(275,60\right)30^{\circ }$> is suggested.

## 4. Conclusions

In this study, we proposed a novel scheme by applying two piecewise cubic Ferguson segments to design variable AFP trajectories, illustrated the effects of parameter settings on variable stiffness laminates, and provided the results of numerous numerical experiments on trajectory design utilizing finite element analysis. The proposed method can modify the buckling load by adjusting the introducing parameters α and θ. A class of reasonable parameter sequences can be applied to design a variable stiffness laminate with a higher buckling load and a smaller curvature. We also suggested a class of optimal parameter sequences <$30^{\circ }\left(50,275\right)90^{\circ }\left(275,50\right)30^{\circ }$>. In future studies, we will explore the use of optimal algorithms, such as deep learning and genetic algorithms, to automatically obtain a set of optimal parameter sequences.

## Figures and Tables

**Figure 1 materials-16-06866-f001:**
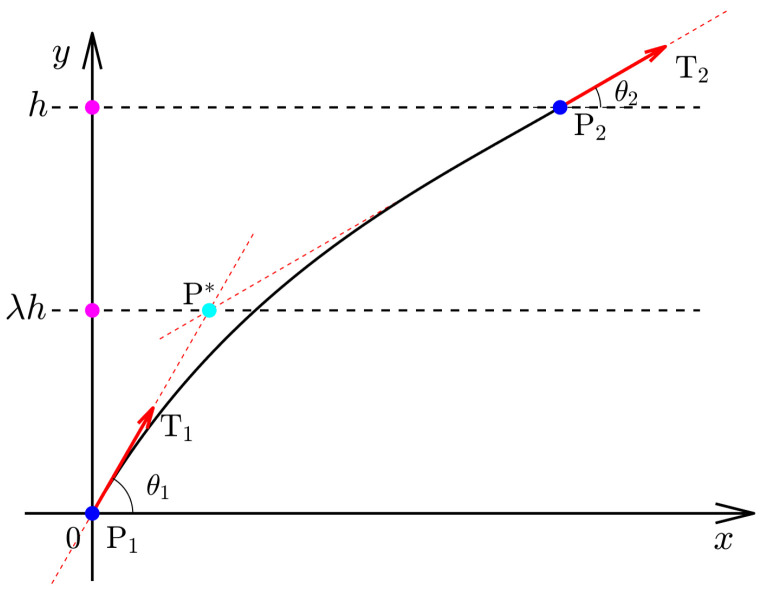
Principle of the AFP trajectory designed by the cubic Ferguson curve.

**Figure 2 materials-16-06866-f002:**
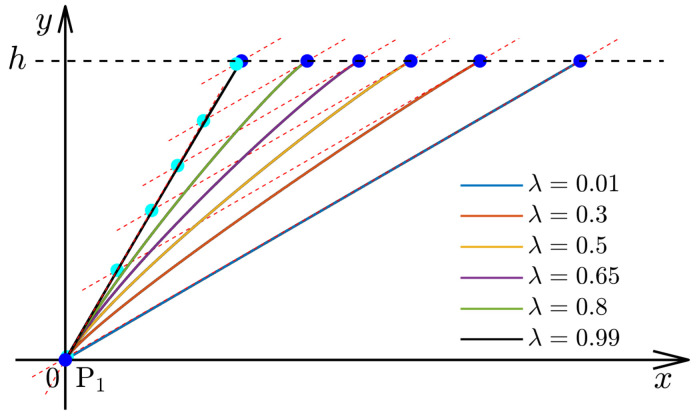
Effects of parameter λ on Ferguson curves. Similar to Figure 1, the blue points are the endpoints $P_{2}$ of the Ferguson curves and the cyan points are $P^{*}$. In this case, we have h=3, $\alpha_{1}=\alpha_{2}=5$, $\theta_{1}=60^{\circ }$, and $\theta_{2}=30^{\circ }$.

**Figure 3 materials-16-06866-f003:**
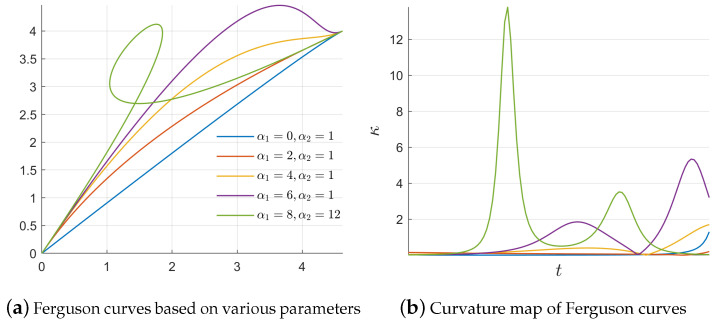
Local adjustment of Ferguson curves relative to parameter $\alpha_{1}$ and $\alpha_{2}$. In this case, we have λ=0.5, h=4, $\theta_{1}=60^{\circ }$, and $\theta_{2}=30^{\circ }$.

**Figure 4 materials-16-06866-f004:**
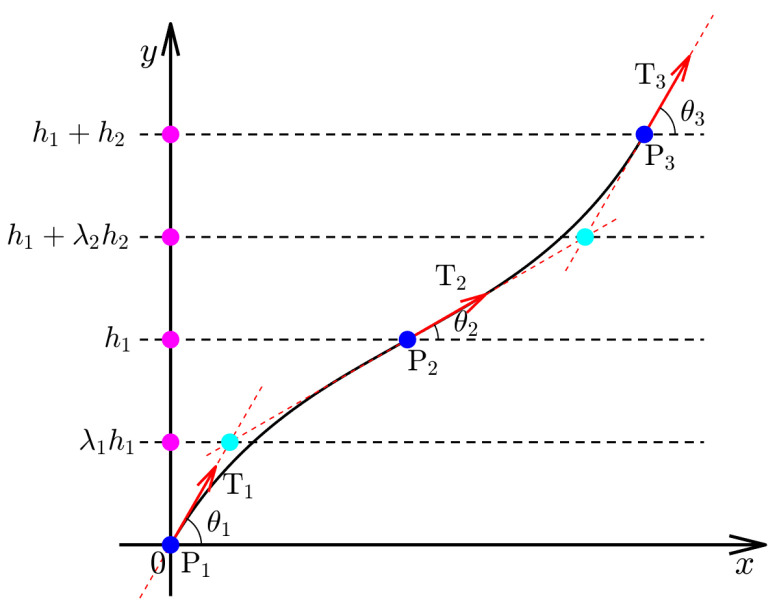
Connection of two AFP trajectory segments. Blue points are knots P and red arrows are vectors T.

**Figure 5 materials-16-06866-f005:**
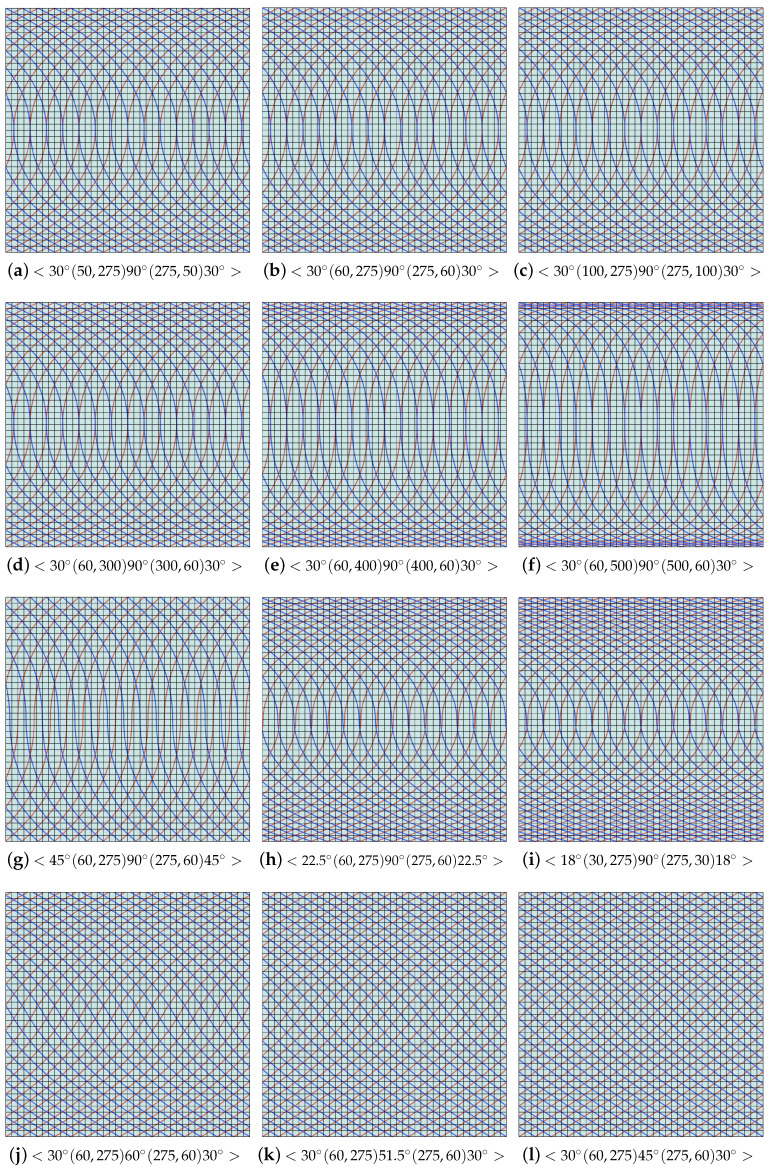
Full cover trajectories with different parameters. In this case, the red (Θ) and blue ($\frac{\pi }{2}-\Theta$) curves represent full cover trajectories over two adjacent plies.

**Figure 6 materials-16-06866-f006:**
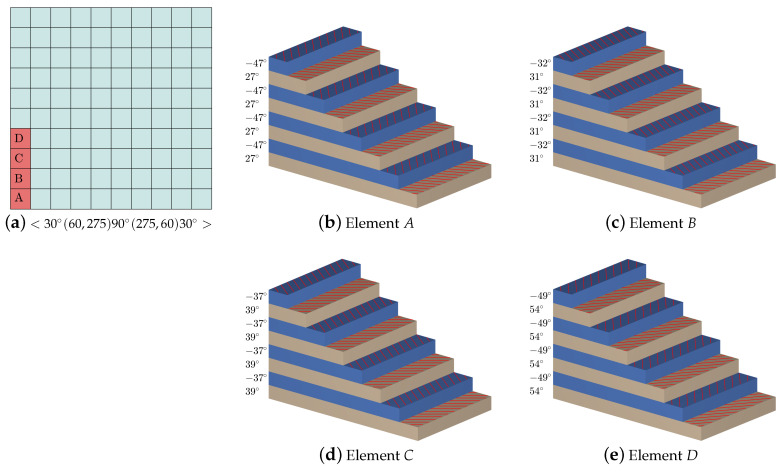
Fiber direction of different layers at the same elements *A*, *B*, *C*, and *D*.

**Figure 7 materials-16-06866-f007:**
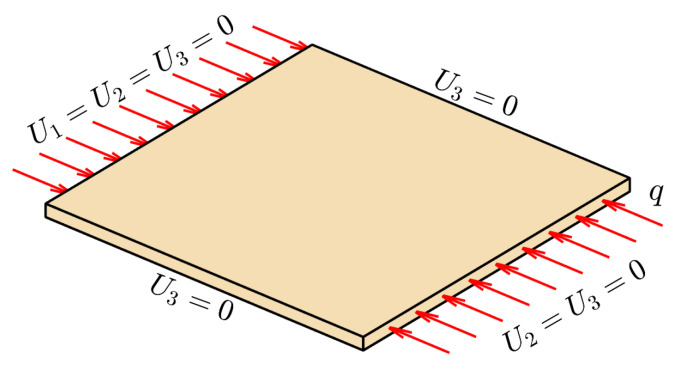
Essential boundary and Neiman boundary conditions of simulated composite laminates.

**Figure 8 materials-16-06866-f008:**
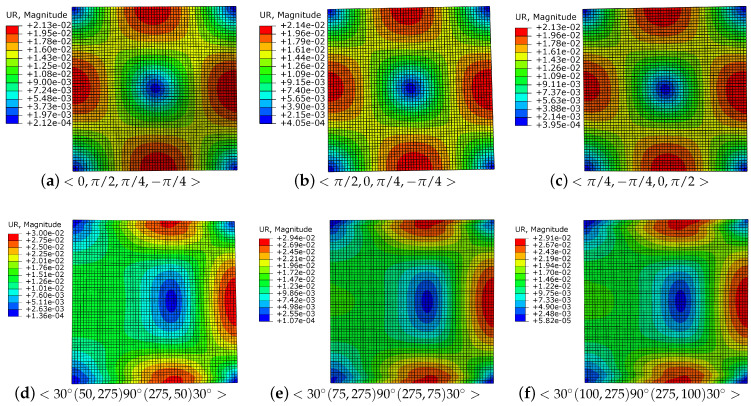
Full cover trajectories with different parameters $\alpha_{1}$, $\alpha_{2}$, and $\alpha_{3}$.

**Figure 9 materials-16-06866-f009:**
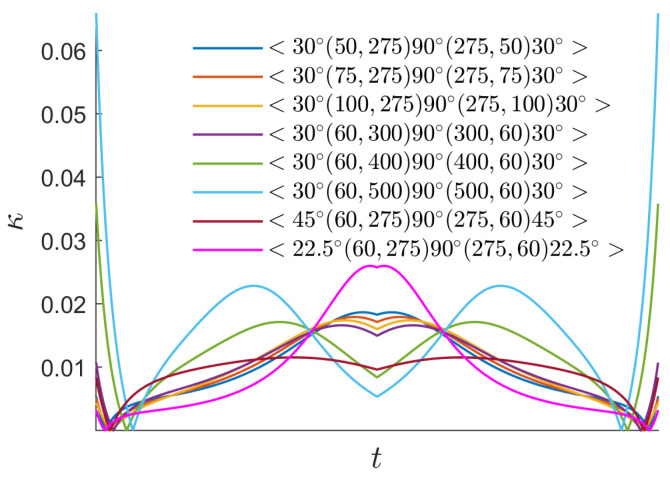
Curvatures κ of initial AFP trajectories relative to parameter *t*.

**Table 1 materials-16-06866-t001:** Elastic foundation parameters of carbon fiber-reinforced towpreg.

$E_{1}/\mathrm{GPa}$	$E_{2}/\mathrm{GP}a$	$G_{12}/\mathrm{GP}a$	$G_{13}/\mathrm{GP}a$	$G_{23}/\mathrm{GP}a$	$\mu_{12}$
120.45	9.51	5.10	5.10	3.63	0.32

**Table 2 materials-16-06866-t002:** Buckling results of composite laminates under different parameter sequences.

Laminate Types	Parameter Settings/θ	Eigenvalues	Buckling Load/N	Maximum Curvature
Constant	<0,90,45,−45>	8.1055	1215.8	0
	<90,0,45,−45>	8.0545	1208.2	0
	<45,−45,0,90>	8.0545	1208.2	0
	<45,−45,90,0>	8.1055	1215.8	0
	<0,45,90,−45>	7.6601	1149.0	0
	<0,−45,90,45>	7.6601	1149.0	0
	<0,45,−45,90>	7.2466	1087.0	0
	<0,−45,45,90>	7.2466	1087.0	0
	<0,90,0,90>	6.4266	963.9	0
	<45,−45,45,−45>	8.1781	1226.7	0
	<90,0,90,0>	6.4266	963.9	0
	<−45,45,−45,45>	8.1781	1226.7	0
Variable	<$30^{\circ }\left(50,275\right)90^{\circ }\left(275,50\right)30^{\circ }$>	10.563	1584.5	0.0344
	<$30^{\circ }\left(60,275\right)90^{\circ }\left(275,60\right)30^{\circ }$>	10.452	1567.8	0.0269
	<$30^{\circ }\left(75,275\right)90^{\circ }\left(275,75\right)30^{\circ }$>	10.297	1544.6	0.0194
	<$30^{\circ }\left(100,275\right)90^{\circ }\left(275,100\right)30^{\circ }$>	10.088	1513.2	0.0174
	<$30^{\circ }\left(150,275\right)90^{\circ }\left(275,150\right)30^{\circ }$>	9.8124	1471.9	0.0171
	<$30^{\circ }\left(60,275\right)90^{\circ }\left(300,60\right)30^{\circ }$>	10.435	1565.3	0.0422
	<$30^{\circ }\left(60,275\right)90^{\circ }\left(400,60\right)30^{\circ }$>	9.6106	1441.3	0.1082
	<$30^{\circ }\left(60,275\right)90^{\circ }\left(500,60\right)30^{\circ }$>	7.6440	1146.6	0.1819
	<$45^{\circ }\left(60,275\right)90^{\circ }\left(275,150\right)60^{\circ }$>	9.1404	1371.1	0.0155
	<$36^{\circ }\left(60,275\right)90^{\circ }\left(275,150\right)60^{\circ }$>	8.8989	1334.8	0.0103
	<$26^{\circ }\left(60,275\right)90^{\circ }\left(275,150\right)60^{\circ }$>	8.6832	1302.5	0.0075
	<$22.5^{\circ }\left(60,275\right)90^{\circ }\left(275,150\right)60^{\circ }$>	8.5022	1275.3	0.0078
	<$20^{\circ }\left(60,275\right)90^{\circ }\left(275,60\right)20^{\circ }$>	9.9857	1497.9	0.0268
	<$18^{\circ }\left(60,275\right)90^{\circ }\left(275,60\right)18^{\circ }$>	10.865	1628.8	0.0277
	<$30^{\circ }\left(60,275\right)72^{\circ }\left(275,60\right)30^{\circ }$>	9.7859	1467.9	0.0253
	<$30^{\circ }\left(60,275\right)60^{\circ }\left(275,60\right)30^{\circ }$>	9.1795	1376.9	0.0260
	<$30^{\circ }\left(60,275\right)52^{\circ }\left(275,60\right)30^{\circ }$>	8.7090	1309.4	0.0260
	<$30^{\circ }\left(60,275\right)45^{\circ }\left(275,60\right)30^{\circ }$>	8.3296	1249.4	0.0299
	<$60^{\circ }\left(60,275\right)18^{\circ }\left(275,60\right)60^{\circ }$>	6.5797	1140.1	0.1644
	<$30^{\circ }\left(60,275\right)22.5^{\circ }\left(275,60\right)30^{\circ }$>	7.6204	1143.1	0.1179
	<$30^{\circ }\left(60,275\right)26^{\circ }\left(275,60\right)30^{\circ }$>	7.6121	1141.8	0.0911
	<$30^{\circ }\left(60,275\right)30^{\circ }\left(275,60\right)30^{\circ }$>	7.6028	1140.4	0.0628
	<$30^{\circ }\left(60,275\right)36^{\circ }\left(275,60\right)30^{\circ }$>	7.6111	1141.7	0.0345
	<$60^{\circ }\left(60,275\right)45^{\circ }\left(275,60\right)60^{\circ }$>	7.6477	1147.2	0.0101

## Data Availability

Not applicable.

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
