# Peer review of "Design of Variable Stiffness Trajectories with Cubic Ferguson Curve"

_materials, 2023, doi:10.3390/ma16216866_

Round 1

Reviewer 1 Report

In order to satisfy curvature limitations and enhance the buckling load of constructed laminates, the paper proposes a novel method by applying two piecewise cubic Ferguson segments to design variable AFP trajectories. The study illustrates the effects of parameter settings on variable-stiffness laminates. The proposed method can modify the buckling load by adjusting the introducing parameters α and θ. After analysing the research objectives and the results, the reviewer considers that the paper should be improved in order to be published in the journal Materials, as follows:

-        within the chapter Definition of variable stiffness laminates the relations must have a citing source and have to be numbered;

-        the authors should also perform some experimental tests (not only FEA);

Author Response

Comments 1: within the chapter Definition of variable stiffness laminates the relations must have a citing source and have to be numbered;

Answer: The related citing sources are provided.

Comments 2: the authors should also perform some experimental tests (not only FEA);

Answer: Our main contribution is that we provide a novel scheme using the cubic Ferguson curve to design variable stiffness laminates. The proposed method can enhance the buckling load by locally modifying the introduced parameters. However, developing software for fiber placement needs to take a long time. So far, there is no large equipment and necessary financial support, and it cannot be achieved in the short term. Therefore, the extra experiments are not supported. Moreover, we believe that the experiments for FEA is enough for showing the efficiency of provided method. Please allow us not to add extra experiments.

Reviewer 2 Report

The study is interesting and the reviewer considers that the manuscript can be published in this journal.

However there are some points that I suggest to be considered:

- Include the effect of the ticknness of each layers of the laminate. 

- The influence of the composite thickness of the composite laminate. 

- include also the unidirectional composite plate 0º and 90º and compare with angled composite plates. 

Author Response

Comments 1: Include the effect of the ticknness of each layers of the laminate. 

Response 1: Firstly, we greatly appreciate your valuable comment. Our main contribution is that we provide a scheme using the cubic Ferguson curve to design variable stiffness laminates. We mainly consider the influence of different fiber trajectory on the variable stiffness laminates. Actually, its core is to design fiber direction with cubic Ferguson curve. Therefore, the effect of the ticknness of each layers of the laminate is independent of our paper. Therefore, we decide that we will not provide the effect of the ticknness of each layers of the laminate. Please allow us to do this.

Comments 2: The influence of the composite thickness of the composite laminate. 

Response 2: We also do not consider the influence of the composite thickness of the composite laminate. The reason is same with the comment 1. Please allow us to do this .

Comments 3: include also the unidirectional composite plate 0º and 90º and compare with angled composite plates. 

Response 3: We have added the undirectional composite plate 0º and 90º on the table 2.

Reviewer 3 Report

The article presents the mathematical model and numerical validation of variable stiffness laminate using the cubic Ferguson curve to improve the buckling load of constructed laminates. Conventional cubic Ferguson curve in polar coordinates is refined and two more segments are introduced with three additional parameters to explain the complete coverage trajectories. The finite element method is used to analyze the bucking load of these variable stiffness laminates. Design of variable stiffness laminate with higher buckling load and small curvature is done with optimal parameter sequence <30 (60, 275)90(275, 60)90 >.  The study contributes to the design of the variable composite laminate when buckling load and curvature are concerned for the application.

Minor Revisions:

1.  Rewrite the heading number, start the introduction from 1, and so on.

2. Re-draw Figure 1 or make another figure in which T0 and T3 vectors are mentioned with α1 and α2.

3.  Re-write the Figure 7 (a) caption.

4.  Please add references to all equations used in this paper. If derived other than the Ferguson curve, mention the reference for lines 84,93,95,96,98,114,127,130,137 equations if taken from literature.

5.  Write down the x-axis and y-axis Units and what these values show on both the horizontal and vertical axis.

 Major Revisions:

Introduction

1.  Add more literature review and background study from recent year's literature.

2.  Why you have selected the Ferguson curve, does this modification give maximum buckling for the variable stiffness composite? Are there any other models used to get the same objective as mentioned in your research article?

3.  What are λ1 and λ2 in Figure 2, and how their value is selected?

Definition of variable stiffness laminates

4. To obtain the ideal trajectory, the magnitude of α1 and α2 should be less than three times the difference between the two endpoints. On what assumption this relationship is created?  

5.   Modify the figure 2, showing the λ close to 0 and 1. Also, in Figure 2, there are variable λ values mentioned in Figure 2. You have λ1 = λ2 = 5 for your case, clarify for which case you are explaining in Figure 2.

6.  Add some explanation as to why the curvature increased as α1 and α2 values are increased. Add some explanation under Figure 3, explaining the effect of α1 and α2 as these parameters are key parameters of your study.

7. For lines 131 – 141, Can you make a general figure that can show all the parameters for a better understanding of θk and θk+1, αk, αk+1, P1, P2, P3.

Analyses and discussion

8.  This section needs more clarification. It is suggested to rewrite with explaining the optimal parameter sequence results.

9. Why this < 30◦ (60, 275)90◦ (275, 60)90◦ > is selected as the optimal sequence as the buckling load is less than this parameter setting < 30◦ (50, 275)90◦ (275, 60)90◦ >?

10. Figure 8, the explanation is required justifying the selected optimal sequence.

Conclusion

11. The portion of the paper is written very clearly and exceptionally well written. Just add what were your numerical analysis contribution and conclusions. 

minor editing required

Author Response

Minor Revisions:

Comments 1:  Rewrite the heading number, start the introduction from 1, and so on.

Response 1: We have rewritten the heading number.

Comments 2: Re-draw Figure 1 or make another figure in which T0 and T3 vectors are mentioned with α1 and α2.

Response 2: From the definition of variable stiffness laminates, the two vectors T0 and T3 are independent with α1 and α2. Therefore, we have not re-drawn Figure 1. Please allow us to do this.

Comments 3:  Re-write the Figure 7 (a) caption.

Response 3: The caption of the figure 7(a) has been rewritten.

Comments 4:  Please add references to all equations used in this paper. If derived other than the Ferguson curve, mention the reference for lines 84,93,95,96,98,114,127,130,137 equations if taken from literature.

Response 4: The references for all equations have been added.

Comments 5:  Write down the x-axis and y-axis Units and what these values show on both the horizontal and vertical axis.

Response 5: The x-axis and y-axis are provided in ome necessary figures, such as figure 2 and 6.

Major Revisions:

Introduction

Comments 1:  Add more literature review and background study from recent year's literature.

Response 1: Some extra literature can be added in Section 1.

Comments 2: Why you have selected the Ferguson curve, does this modification give maximum buckling for the variable stiffness composite? Are there any other models used to get the same objective as mentioned in your research article?

Response 2: Some advantages of the Ferguson curve have been provided in Section 1 and labeled in red.

Comments 3:  What are λ1 and λ2 in Figure 2, and how their value is selected?

Response 3: This is a small mistake and we have modified this mistake.

Definition of variable stiffness laminates

Comments 4: To obtain the ideal trajectory, the magnitude of α1 and α2 should be less than three times the difference between the two endpoints. On what assumption this relationship is created?  

Response 4: We have provided some explanations and key literatures in subsection 2.2.

Comments 5:  Modify the figure 2, showing the λ close to 0 and 1. Also, in Figure 2, there are variable λ values mentioned in Figure 2. You have λ1 = λ2 = 5 for your case, clarify for which case you are explaining in Figure 2.

Response 5: This is a small mistake and we have rewritten this description in cation of Figure 2.

Comments 6:  Add some explanation as to why the curvature increased as α1 and α2 values are increased. Add some explanation under Figure 3, explaining the effect of α1 and α2 as these parameters are key parameters of your study.

Response 6: The influence of parameters $\alpha_1$ and $\alpha_2$ on cubic Ferguson curves and the determination of the interval magnitude is not the main work of this paper and the related work is provided. Please allow us to do this.

Comments 7: For lines 131 – 141, Can you make a general figure that can show all the parameters for a better understanding of θk and θk+1, αk, αk+1, P1, P2, P3.

Response 7: We think that the redraw figure 4 can be used to show the relations of parameters  θk , θk+1, αk, αk+1, P1, P2, P3. Therefore, we have not provided an extra one. Please allow us to do this.

Analyses and discussion

Comments 8:  This section needs more clarification. It is suggested to rewrite with explaining the optimal parameter sequence results.

Response 8: The explaining for the optimal parameter sequence has been provided and labeled in red.

Comments 9: Why this < 30◦ (60, 275)90◦ (275, 60)90◦ > is selected as the optimal sequence as the buckling load is less than this parameter setting < 30◦ (50, 275)90◦ (275, 60)90◦ >?

Response 9: A practical optimal parameter sequence is needed to consider both of curvature limitation and buckling load. Therefore, the parameter setting < 30◦ (50, 275)90◦ (275, 60)90◦ > is suggested.

Comments 10: Figure 8, the explanation is required justifying the selected optimal sequence.

Response 10: The Figure 8 shows full cover trajectories with different parameters on a laminate. Its related explanation has been provided in Section 3 labeled in red.

Conclusion

Comments 11: The portion of the paper is written very clearly and exceptionally well written. Just add what were your numerical analysis contribution and conclusions. 

Response 11: The numerical analysis contribution and conclusions have been added.

Reviewer 4 Report

Dear Authors,

in my opinion, the manuscript you have prepared can be of high potential interest to readers, due to the innovative approach to layered laminate materials design and production, including automated fibre placement (AFP) according to the distribution given by the cubic Ferguson curve.

The manuscript is well prepared. I have just two remarks:

- line 15 – instead of “radio” word in my opinion there should be “ratio”, but I’m not sure what kind of “ratio”

- figure 6 – I suggest adding the X and Y axis descriptions and units

Best regards!

Author Response

Comments 1: line 15 – instead of “radio” word in my opinion there should be “ratio”, but I’m not sure what kind of “ratio”

Response 1: Answer: I have replaced the word ‘radio’ by ‘ratio’.

Comments 2: figure 6 – I suggest adding the X and Y axis descriptions and units

Response 2: The X and Y axis has been added in Figure 3,6.